# A Morphing Point-to-Point Displacement Control Based on Long Short-Term Memory for a Coplanar XXY Stage

**DOI:** 10.3390/s23041938

**Published:** 2023-02-09

**Authors:** Ming-Yu Ma, Yi-Cheng Huang, Yu-Tso Wu

**Affiliations:** 1Department of Mechanical Engineering, National Chung Hsing University, Taichung City 40227, Taiwan; 2Department of Mechatronics Engineering, National Changhua University of Education, Changhua City 500207, Taiwan

**Keywords:** feedback control, image recognition, long short-term memory, XXY stage

## Abstract

In this study, visual recognition with a charge-coupled device (CCD) image feedback control system was used to record the movement of a coplanar XXY stage. The position of the stage is fedback through the image positioning method, and the positioning compensation of the stage is performed by the image compensation control parameter. The image resolution was constrained and resulted in an average positioning error of the optimized control parameter of 6.712 µm, with the root mean square error being 2.802 µm, and the settling time being approximately 7 s. The merit of a long short-term memory (LSTM) deep learning model is that it can identify long-term dependencies and sequential state data to determine the next control signal. As for improving the positioning performance, LSTM was used to develop a training model for stage motion with an additional dial indicator with an accuracy of 1 μm being used to record the XXY position information. After removing the assisting dial indicator, a new LSTM-based XXY feedback control system was subsequently constructed to reduce the positioning error. In other words, the morphing control signals are dependent not only on time, but also on the iterations of the LSTM learning process. Point-to-point commanded forward, backward and repeated back-and-forth repetitive motions were conducted. Experimental results revealed that the average positioning error achieved after using the LSTM model was 2.085 µm, with the root mean square error being 2.681 µm, and a settling time of 2.02 s. With the assistance of LSTM, the stage exhibited a higher control accuracy and less settling time than did the CCD imaging system according to three positioning indices.

## 1. Introduction

With the rapid development of industrial science and technology, products are becoming smaller and more complex. High-precision requirements are particularly crucial for improving assembly and reducing the need for inspection. Long working hours can easily lead to worker fatigue, and different workers have a different production accuracy. Therefore, product accuracy is difficult to maintain. In recent years, machine vision [1] has been widely used in auxiliary identification. The images are captured by a charge-coupled device (CCD) camera, image processing, and identification to achieve a consistency of identification results.

Industry 4.0 is oriented toward microelectromechanical and micro–nano systems, and relevant product development aims at lightness and portability. Therefore, semiconductor processing equipment and machine tools will develop in this direction. To meet the requirements for producing miniature workpieces, the machining accuracy demands of the machine tool industry have gradually increased. The precision positioning stage can enable the accuracy demands for miniature machining to be satisfied.

Geometrical alignment in stacked XYθz stages usually causes cumulative errors, such as in parallelism, orthogonal between two axes, and flatness errors. In [2,3], the devised coplanar XYθz stages were used to avoid such errors. They exhibited superior positioning accuracy compared with the stacked type stages. In [4], the feature of alignment was performed in wafer manufacturing by using two CCD cameras with an XYθz stage and a feed-forward neural network controller. The positioning precision was approximately 12 μm. The experiment employed a cross mark for alignment. Cross marks are commonly used in many visual alignment systems because they have clear features and can be easily identified. In [5], a fuzzy logic controller was used to control the motion of the alignment stacked XY stage with the imaging software eVision for image processing.

In [6], an XXY stage mask alignment system with dual CCD image servos was designed. On the basis of the motion method of an artificial neural network, nonlinear mapping of the stage at the required position was conducted, and the directions of the commands of three motors were established. In [7], the XXY stage was integrated with dual CCDs to establish an automatic alignment and recognition system. Automated optical inspection (AOI) technology was proposed to guide the stage and combine image technology with stage control technology to perform image recognition and alignment tasks. In [8,9], a specially designed coplanar stage for visual servo control and an image alignment system were proposed, and the alignment movement and error of the image on the coplanar XXY stage were analyzed. Subsequently, the influence of kinematics analysis and setting errors were discussed. The image alignment method for a floating reference point was proposed to reduce the effect of the alignment error between the center of the workpiece and the reference point of the stage. In [10], a microcontroller was used for an image-based XXY positioning platform. The positioning error between the XXY stage and the detected object was determined by image recognition technology. Subsequently, the error information could be used for positioning control. In [11], automatic locating and image servo alignment for the touch panels of a laminating machine was employed via four CCD cameras and a coplanar XXY stage also.

A long short-term memory (LSTM) network [12] is a variant of a recurrent neural network (RNN) and was first proposed in 1997. Because of their unique design, LSTM networks are often used to handle time-series data problems to solve the vanishing gradient problem. LSTM networks are complex nonlinear units that can be used to construct a large deep learning network. In recent years, many studies have used LSTM networks to address different problems related to time-series data.

In [13], to ensure that the positioning and navigation systems of intelligent network vehicles can still output a location with high accuracy when global navigation satellite system (GNSS) positioning failure occurs, a high-precision positioning method based on LSTM was proposed. Experimental results indicated that the method’s performance met the high-precision positioning and navigation requirements of intelligent network vehicles on urban roads. The results demonstrated that the LSTM network could approximate the relationship between the input and output of a GNSS-integrated navigation system (INS) with high precision. In [14], a visual recognition system was combined with an artificial-intelligence machine-learning neural network to predict the maximum pick-and-place offset of a robot arm in the next minute. The developed LSTM model made predictions with high accuracy and reliability and met handling robot needs. In [15], the LSTM model predicted the remaining life of gears, and the LSTM networks could capture both short-term correlation and long-term dependence. In [16], a system was proposed for accurately detecting the lane line (LL) on roads to improve vehicle driving safety. An LL prediction model based on LSTM was established according to the spatial information of LLs and the distribution law of LLs, and future LL location was predicted using historical LL location data. In [17], the use of LSTM to improve the PSO algorithm by finding the best fitness value on the XXY stage was conducted with three types of motion. The developed LSTM could predict the fitness value of PSO by eliminating the need to preassess the fitness value, and adjusted the inertia weight of PSO adaptively. The experimental results indicated that LSTM could reduce the time to find optimal control parameters, and the stage positioning error for the XXY stage could be reduced significantly.

In the present study, the positioning information of an XXY micromotion stage was used to construct a predictive model with a time series through machine learning. First, stage displacement data were collected through imaging, and the collected data was used to establish a training dataset as the training and verification subset of the neural network. Subsequently, the LSTM model predicted the next movement. Finally, the stage was adjusted according to the results predicted by the model to achieve the optimal positioning requirements.

The rest of this paper is organized as follows. Section 2 describes the experimental setup, including the XXY stage, vision system, and controller; Section 3 introduces the LSTM network used in this study; Section 4 provides details on the experimental process; Section 5 describes the analysis of the experimental results; and Section 6 provides the conclusions of this study.

## 2. Image Capture System

An XXY stage is characterized by three motors on the same plane and has the merit of a low center of gravity. Thus, the movement speed of an XXY stage is higher than that of a traditional stacked XYθ stage. The XXY stage is small and light. Therefore, the main advantage of an XXY stage is that it exhibits a smaller cumulative error in stage composition than does a traditional stacked stage. Therefore, a coplanar XXY stage is highly popular for applications that require precision motion, such as AOI and lithography.

As illustrated in Figure 1, the experimental system consisted of an upper mask chip device, which carried the upper cross mask for CCD imaging; a lower coplanar XXY stage (XXY-25-7, CHIUAN YAN Ltd., Changhua, Taiwan) [18], which carried the lower part to align the upper device with the image servo control; and two CCD camera lenses, which were mounted on top of the system as the image servo sensors for positioning. A motion card (PCI-8143, ADLINK TECHNOLOGY Inc., Taoyuan, Taiwan) controlled the XXY stage [19], and ADLINK’s Domino Alpha2 image card provided the XXY stage with image position feedback.

Traditional XYθ stages use a stacked design, which consists of an *x*-axis translation stage, a *y*-axis translation stage, and a θ-axis rotational stage. The controller design for a traditional stacked XYθ stage is simple because each axis moves independently. However, the XYθ stage produces cumulative flatness errors because of its stacked assembly and large size. Therefore, a coplanar XXY stage was developed because the coplanar design produces relatively low cumulative errors and can move faster than does the traditional XYθ stage. Figure 2 displays the structure of the coplanar XXY stage, which is driven by three servo motors: an x1-axis motor, an x2-axis motor, and a *y*-axis motor. The working stage is supported by four substages, each of which consists of *x*-axis translation, *y*-axis translation, and θ-axis rotation stages. Therefore, the motion of the XXY stage has three degrees of freedom: translation along the *x*-axis and *y*-axis and rotation around the θ-axis. Figure 3 illustrates the *y*-axis movement. Table 1 provides the specifications for the XXY stage used in this study (CHIUAN YAN Ltd.) [18], and Table 2 presents the specifications of the CCD device.

This study used the OpenCV software 4.5 version (Intel Corporation, Mountain View, CA, USA) for image processing. The image of a cross mark captured by the CCD camera was used for image processing to obtain stage positioning. The dimensions of the cross mark were 1 mm × 1 mm. Image processing involved grayscaling, binarization, and erosion and expansion to eliminate noise. Finally, the image center in the gravity method was used to obtain the image coordinates, and the center of gravity coordinates were used to calculate the image displacement.

When the stage was moving, the CCD captured an image and saved it as an image file, and the execution time of the CCD was 0.5 s. Each pixel of the image contained three colors—red, green, and blue. After the image was grayscaled, each pixel was changed to black or white with a brightness between 0 and 255. The brightness was lower for dark colors and higher for light colors. Subsequently, binarization was used to convert the pixel information to 0 (black) or 255 (white). The conversion method depended on the threshold set, and the threshold was 100. If the pixel value was above the threshold, the pixel value was converted to 255. If the pixel value was below the threshold, the pixel value was converted to 0. After the image was binarized, the image only had two colors—black and white—which facilitated image processing (Figure 4).

After the image was binarized, the image had stray white and black dots because of the influence of light and the threshold settings. Erosion processing eliminated the small white dots (Figure 5), and dilation processing eliminated the small black dots (Figure 6). Erosion processing resulted in the conversion of the white pixels around a black pixel into black pixels, and dilation processing resulted in the opposite effect. The aforementioned image processing methods can distort the original image if the range of erosion and dilation is excessively large; therefore, a dilation and erosion range of three pixels was set.

A reference point is required in image positioning. In this study, the reference point was the cross mark. The image center of gravity method was used to find the center of gravity of the cross mark, and the stage displacement was then calculated. This method is a simple method used to find image reference coordinates. The image centroid method was used to find the fiducial coordinates of an image, as shown in Figure 7. In this method, the position coordinates of the pixels of the reference point are added, and the obtained values are divided by the total number of pixels of the reference point to obtain the coordinates of the center of gravity. Subsequently, the stage displacement can be calculated using these coordinates.
(1)$$C_{x}=\frac{\sum X_{C}}{N}$$
(2)$$C_{y}=\frac{\sum Y_{C}}{N}$$

In (1) and (2), $C_{x}$ is the *x*-coordinate of the center of gravity, $C_{y}$ is the *y*-coordinate of the center of gravity, $\sum X_{C}$ is the sum of the *x*-coordinates of the white cross image, $\sum Y_{C}$ is the sum of the *y*-coordinates of the white cross image, and *N* is the sum of the white pixels of the white cross image.

## 3. LSTM Network

LSTM networks are variants of RNNs [20] and inherit the characteristics of RNN models. Because the new memory of an RNN model overwrites the old memory in the recursive layer, the flow of memory cannot be controlled individually, so LSTM networks can control the flow of memory through the gate. As displayed in Figure 8, $X_{t}$ represents the input data of the RNN at the current time point, and $Y_{t-1}$ represents the output of the hidden layer at the previous time point. If the time increases gradually, the generated time sequence becomes excessively long, and the neural network would be unable to learn the information at the beginning of the dataset. This problem is called gradient vanishing or gradient exploding, and the adjusted gradient weight of the RNN model would be excessively large or small. Therefore, training long-term sequences and obtaining model predictions are difficult tasks for an RNN model.

An LSTM network possesses a memory structure that contains memory cells. It adds and memorizes information as a time series progresses, thereby solving the vanishing gradient problem. Figure 9 illustrates the basic structure of an LSTM network. The cell state can be used to store and transmit memory; therefore, the information in this state can be written or deleted. Without external influences, the aforementioned information remains unchanged. The parameter $x_{t}$ represents the input data at time t, and $h_{t-1}$ is the hidden state at time t−1. The cell state at time t−1 is denoted as $c_{t-1}$, which is modified from the present cell state $\;c_{t}$ in the hidden layer at time t.

The hidden layer of an LSTM network contains an input node ($a_{t}$) and three gates ($f_{t},\;i_{t},\;\mathrm{and}\;o_{t}$). The variables $a_{t},f_{t},\;i_{t},\;\mathrm{and}\;o_{t}$ are calculated using (3)–(6), respectively. The input node $a_{t}$ is used for updating the cell state, and the gates are used to determine whether to allow information to pass through. The three gates in an LSTM network are a forget gate, an input gate, and an output gate. The forget gate ($f_{t}$) determines which cell states’ ($c_{t-1}$) information may pass through. The input gate ($i_{t}$) determines which input nodes’ information ($a_{t}$) may pass through. The vectors (information) passing through the input gate are used for updating the cell state and are subjected to element-wise addition by the vectors (information) passing through the forget gate to generate the cell state ($c_{t}$). The calculation in the aforementioned process is expressed in (7). The output gate determines which cell state’s ($c_{t}$) information may pass through it. The vectors (information) passing through the output gate are in the hidden state ($h_{t}$), and they are the output vectors of the current hidden layer. The calculation method for $h_{t}$ is presented in (8). In addition, the cell state and hidden state obtained at time t, namely ($c_{t}$) and ($h_{t}$), respectively, are transmitted to the hidden layer at time t+1. This process, which progresses with a time series, is used for the transmission and learning of memory.
(3)$$i_{t}=\sigma \left(W_{i}x_{t}+H_{i}h_{t-1}+b_{i}\right)$$
(4)$$o_{t}=\sigma \left(W_{o}x_{t}+H_{o}h_{t-1}+b_{o}\right)$$
(5)$$f_{t}=\sigma \left(W_{f}x_{t}+H_{f}h_{t-1}+b_{f}\right)$$
(6)$$a_{t}=\;\tan h\left(W_{a}x_{t}+H_{a}h_{t-1}+b_{a}\right)$$
(7)$$c_{t}=(f_{t}⊙c_{t-1})⊕\left(i_{t}⊙a_{t}\right)$$
(8)$$h_{t}=o_{t}⊙\tan h\left(c_{t}\right)$$
where W and H represent the weight, b denotes the bias, ⊕ is the symbol for element-wise addition, ⊙ is the symbol for element-wise multiplication, tanh denotes the hyperbolic tangent, and σ denotes the sigmoid function. The parameters tanh and σ represent activation functions.

## 4. Experimental Process

The flowchart of the conducted experiment is illustrated in Figure 10. The linear displacement movement of the *y*-axis was controlled between 0 and 1000 µm by using a motion card. The adopted XXY stage had two feedback modes. In the first mode, the dynamic response displacement of the stage was obtained by CCD and OpenCV image processing. In the second mode, the displacement of the stage was measured by a dial indicator as training data, and the stage position was predicted by an LSTM network. Subsequently, the position results, motion trajectory error, and positioning accuracy of the two modes were compared. When the positioning error of the stage predicted by the LSTM model used a dial indicator for the implementation of off-line compensation and lowered the original image feedback positioning, the morphing control command made by the LSTM model for positioning compensation was achieved thereof. Then, mode 1 assisted by adding the LSTM model when mode 2 was tested successfully.

## 5. Experimental Analysis

There were three parts of this experiment. The first was to adjust the KP values in the image positioning system and test the image feedback results. The second was to construct an LSTM model to test three types of motion. Finally, the error and repeatability of the LSTM model were analyzed. The three parts are described below.

### 5.1. Proportional Gain Value of Image Feedback

The dynamic response of the image feedback was different for different proportional gain (KP) values, and the control gain value of the feedback system was the shortest response time in the experiment. As displayed in Figure 11, when KP was 30, the dynamic response was fast and did not overshoot. Overshoot was considered as an experiment condition for two reasons. First, doing so does not make the response time faster or the settling time longer. Second, doing so avoids tool collision accidents during acceleration. Therefore, the KP value (gain value) was set at 30 for image feedback control.

### 5.2. Image Feedback Results

To compare the position error between the nonimage feedback system and the image feedback system, the nonimage feedback system used the encoder for positioning feedback, the image feedback system used CCD for positioning feedback, then the stage was moved to four target positions from the starting point 402. The target positions were 380, 360, 340, and 320. The stage exhibited the same error in short-distance and long-distance movement. After adjusting the image compensation gain value, the position error was reduced to one pixel through image feedback compensation, as displayed in Figure 12 and Figure 13. Although image feedback reduced the position error to one pixel, the position error was still too large because of the limitations of the imaging device. The stage exhibited linear back-and-forth movement and the movement presented a time series relationship. In this study, an LSTM network was used as the positioning prediction model. To improve positioning performance, the LSTM network trained the positioning of the stage each time and then predicted the next displacement.

### 5.3. Construction of the LSTM Network

The LSTM model was implemented using Python’s Keras library, and the training data were the data collected by the dial indicator. We used an LSTM model with a many-to-many architecture, and the data structure was five inputs and five outputs. The data from the current five time series were the input of the neural network [$y_{\left(t\right)}$]. The series data were [$x_{\left(t-4\right)},\;x_{\left(t-3\right)},\;x_{\left(t-2\right)},\;x_{\left(t-1\right)},\;x_{\left(t\right)}$], and the LSTM model predicted the next five steps of data [${y^{\prime }}_{\left(t+1\right)}$]. The predicted data were [$x_{\left(t+1\right)},\;x_{\left(t+2\right)},\;x_{\left(t+3\right)},\;x_{\left(t+4\right)},\;x_{\left(t+5\right)}$]. To analyze the forward, backward, and return modes individually, an LSTM model with a five-to-five architecture was used. The loss function was the mean square error (MSE), and the purpose was to calculate the minimum difference between the predicted position and the actual position to achieve the optimal prediction. The LSTM model contained a single layer; the number of epochs was 200; the batch size was 6; the number of neurons was 30; and the adaptive moment estimation (Adam) optimizer was used. Figure 14 presents the many-to-many architecture of the adopted LSTM network, and Figure 15 presents the LSTM-based loss function graph. The final training MSE was 0.0130, and the validation MSE was 0.0112.

### 5.4. Neural Network Prediction of the Forward Position of the Stage

The first prediction experiment involved obtaining the error for forward motion prediction. The forward motion commands for actual movement were the input data, and the position of the next movement was obtained using the LSTM network; the forward-to- forward network architecture is illustrated in Figure 16. In Figure 17, the left red dashed frame represents the motion command and actual position from 100 to 500 μm, and the right red dashed frame represents the predicted position of the neural network. The five positions are between 600 and 1000 μm. Therefore, the first five forward movements were used to predict the last five forward movements by using the LSTM network. The network accurately predicted the position of the forward movement, and the maximum error for forward motion prediction was 4.880 μm (Figure 17).

### 5.5. Neural Network Prediction of the Backward Position of the Stage

The second prediction experiment involved obtaining the error for backward motion prediction. The backward motion commands for actual movement were the input data, and the position of the next movement was obtained using the LSTM network; the backward-to-backward network architecture is illustrated in Figure 18. In Figure 19, the left red dashed frame represents the motion command and actual position from 1000 to 600 μm, and the right red dashed frame represents the predicted position of the neural network. The five positions were between 500 and 100 μm. Therefore, the first five backward movements were used to predict the last five backward movements by using the LSTM network. The network accurately predicted the position of the backward movement, and the maximum error for backward motion prediction was 2.115 μm.

### 5.6. Neural Network Prediction of the Forward and Backward Positions

The third prediction experiment involved obtaining the error for forward and backward motion prediction. This experiment involved two modes of motion. The back-and-forth (return) motion for actual movement were the input data first, and then the position of the next consecutive backward movement was predicted via the LSTM network. The architecture of back-and-forth followed by backward motion was illustrated in Figure 20. The input data for the first mode are the data in the left red dashed frame in Figure 21, including the return position. The left red dashed frame was the motion command and actual position from 800 to 1000 μm, and then returned to 800 μm. Therefore, the right red dashed frame represents the position predicted by the LSTM network, and the five commanded position ranged from 700 to 300 μm, as illustrated in Figure 21 (top). The second mode was conducted in the reverse movement starting from the position of 300 μm. The input data were the actual forward movement, then the position of the next back-and-forth (return) movement was predicted via the LSTM network. The forward followed by the return network architecture is illustrated in Figure 22. The actual forward movements were the input data, and the position of the next back-and-forth movement was obtained via the LSTM network. The architecture of forward followed by the back-and-forth return motion is illustrated in Figure 22. The neural network predicted the return movement of the stage, ranging from 800 to 1000 μm, as indicated in Figure 23 (top). As the aim for a point-to-point control, Figure 21 and Figure 23 (bottom) demonstrated the successful prediction of a stage repetitive motion. The LSTM neural network exhibited favorable prediction results with maximum errors between 2.470 and 5.937 μm.

### 5.7. Comparison of Position Predicted by the LSTM Network and the Actual Position

After the three motion prediction experiments, the trend of the neural network prediction error curve was similar to the actual error curve. The maximum error was 7.717 μm at the 10th position of the error curve. Furthermore, the difference between the predicted error and the actual error gradually reduced toward the last position, and the final average error was 2.085 μm (Figure 24).

### 5.8. Stage Repetitive Positioning by LSTM

On the basis of the prediction accuracy of the LSTM network, the repeatability of the prediction of this network at the same position was analyzed. To evaluate the repeatability of the neural network, the trend of the predicted position when the stage moved backward to a certain point was determined. Figure 25 depicts the actual and predicted positions of the stage for backward stage movement.

For the 900 μm position, the first and second predicted positions were located marginally above the actual position. The difference between the first prediction and the actual position was 2.938 μm, and the difference between the second prediction and the actual position was 3.107 μm (Figure 26). For the 600 μm position, the positions predicted by the neural network were located below the actual position. The difference between the first prediction and the actual position was −3.742 μm, and the difference between the second prediction and the actual position was −3.558 μm (Figure 27). For the 400 μm position, the predicted positions were located marginally above than the actual position. The difference between the first prediction and the actual position was 0.620 μm, and the difference between the second prediction and the actual position was 0.635 μm (Figure 28). Thus, the predictions of the neural network for the same position exhibited favorable repeatability in the three experiments.

## 6. Conclusions

In this study, an LSTM model was used to construct an XXY stage positioning prediction system. By using the XXY stage’s image feedback compensation system, the positioning error was reduced to one pixel. The image compensation was limited by the imaging equipment used. Therefore, a dial indicator and an LSTM network were used to construct a positioning prediction model. The maximum positioning error was 7.717 μm, the average error was 2.085 μm, and the root MSE was 2.681 μm. The LSTM network exhibited favorable repeatability. Moreover, relatively small positive and negative errors were observed when using the LSTM network, and the trend of the neural network prediction error curve was similar to the actual error curve. The experimental results indicated that an LSTM model can be used for various types of motion positioning prediction, such as forward, backward, and return motion prediction, for an XXY stage. An error was observed between the actual displacement of the stage and the feedback displacement of the encoder. Therefore, an LSTM model with a time-series relationship was established using actual movement information collected by the dial indicator. This model can be used for displacement compensation in control systems for XXY stages.

## Figures and Tables

**Figure 1 sensors-23-01938-f001:**
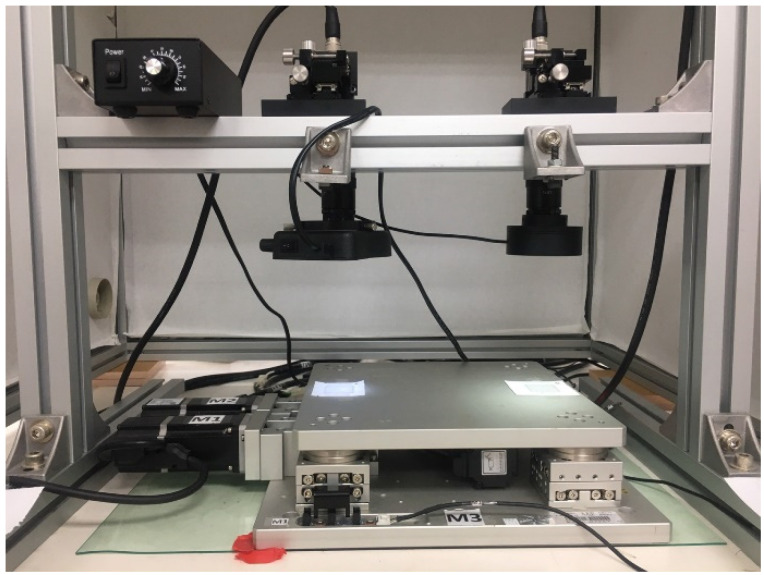
XXY stage and CCD camera.

**Figure 2 sensors-23-01938-f002:**
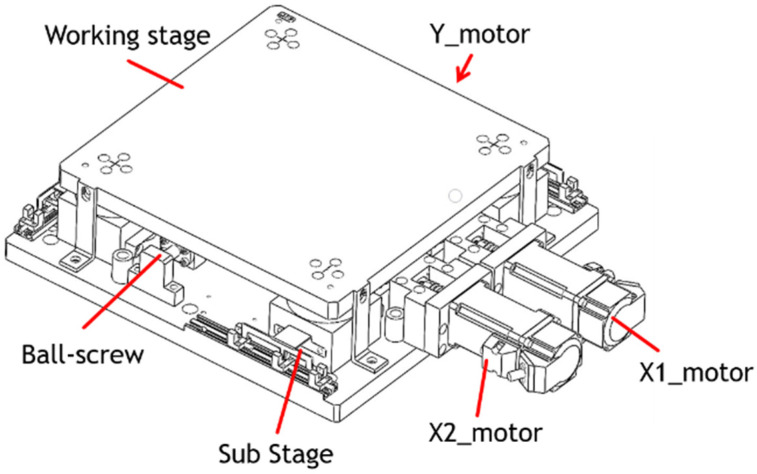
Coplanar XXY stage.

**Figure 3 sensors-23-01938-f003:**
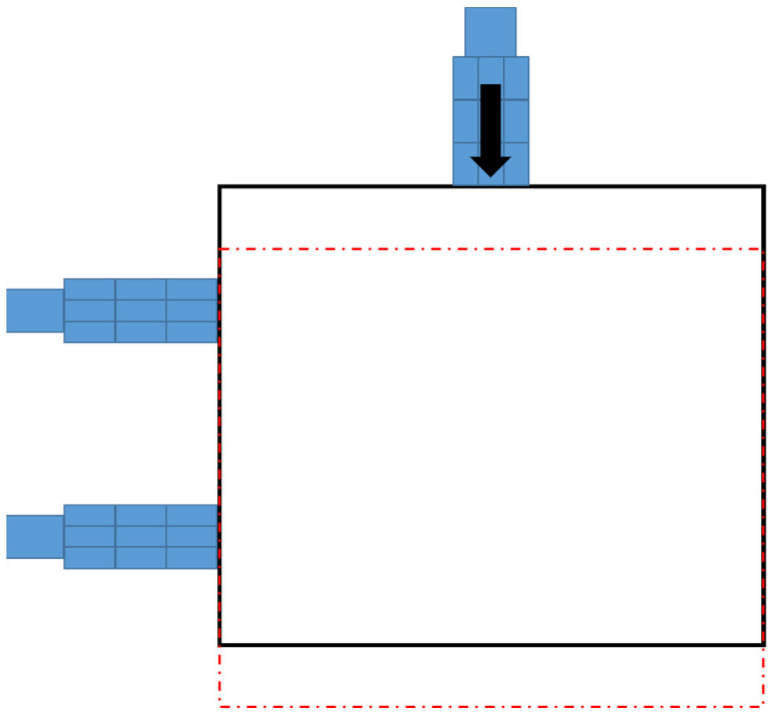
Y-axis movement indicated as red dotted box of the XXY stage.

**Figure 4 sensors-23-01938-f004:**
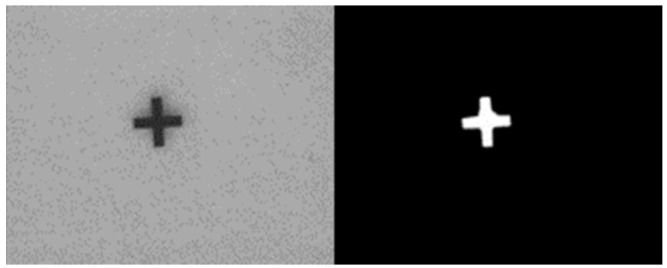
Image gray Ok scaling and binarization of the cross mark for reference.

**Figure 5 sensors-23-01938-f005:**
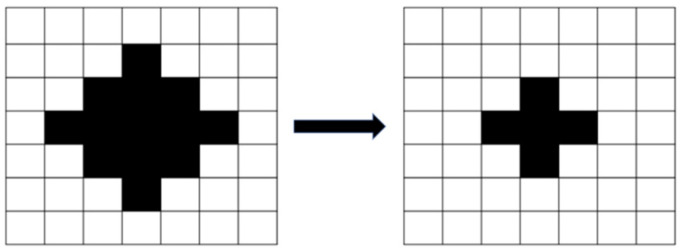
Image erosion.

**Figure 6 sensors-23-01938-f006:**
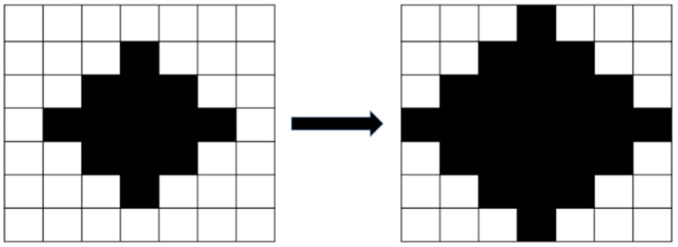
Image dilation.

**Figure 7 sensors-23-01938-f007:**
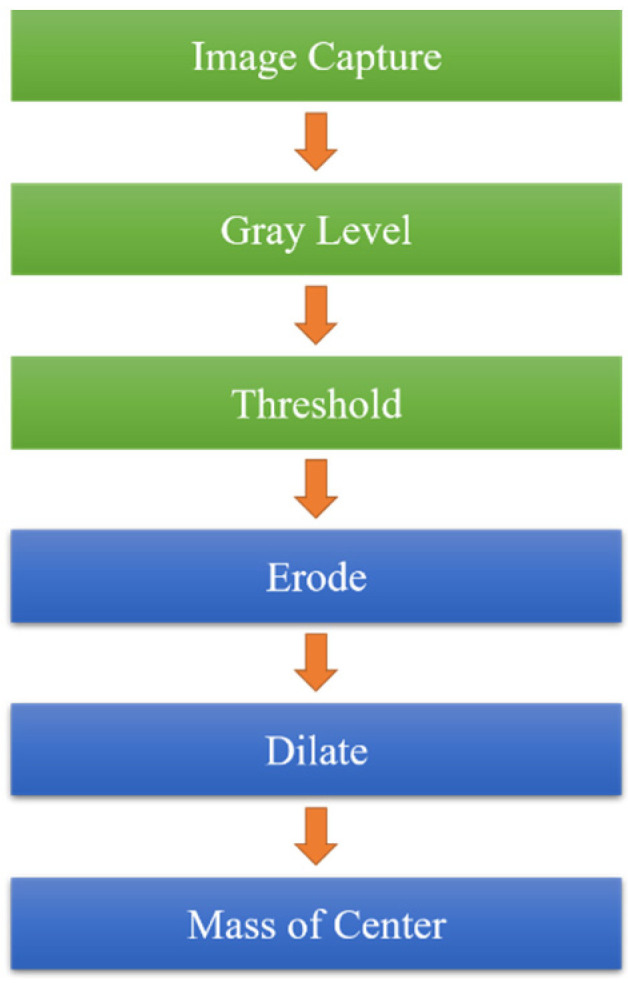
Image processing flow.

**Figure 8 sensors-23-01938-f008:**
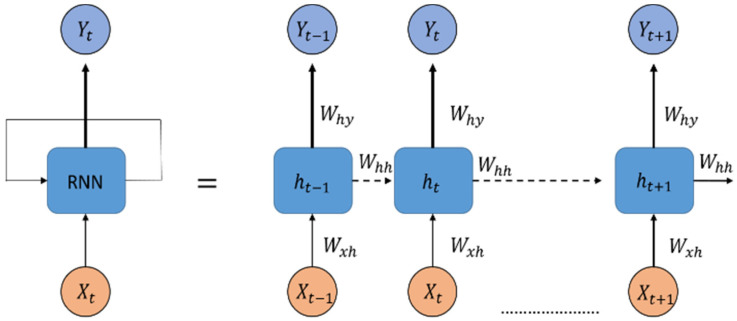
Structure of an RNN.

**Figure 9 sensors-23-01938-f009:**
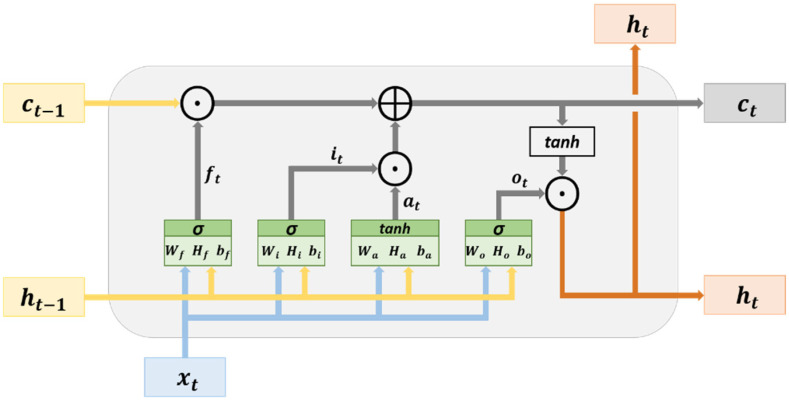
Architecture of an LSTM network.

**Figure 10 sensors-23-01938-f010:**
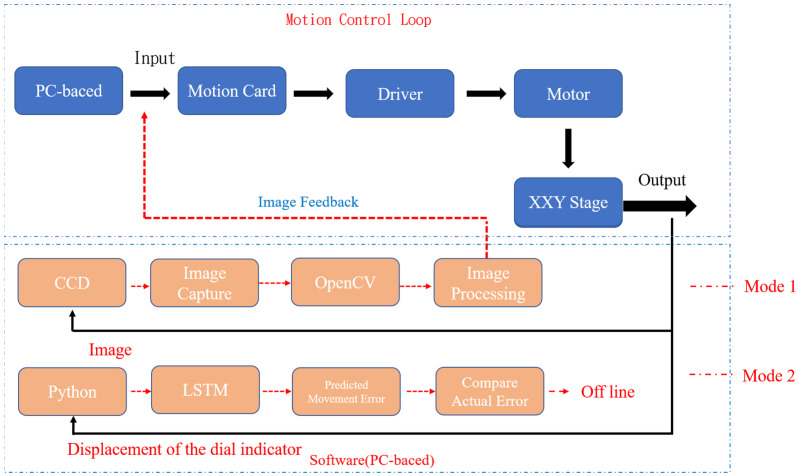
Flowchart of the experiment.

**Figure 11 sensors-23-01938-f011:**
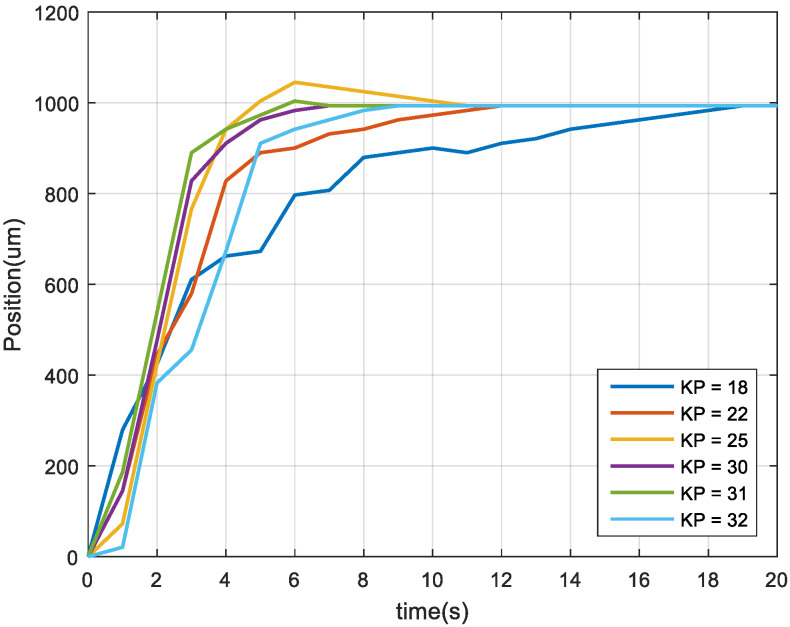
Dynamic responses under different KP values.

**Figure 12 sensors-23-01938-f012:**
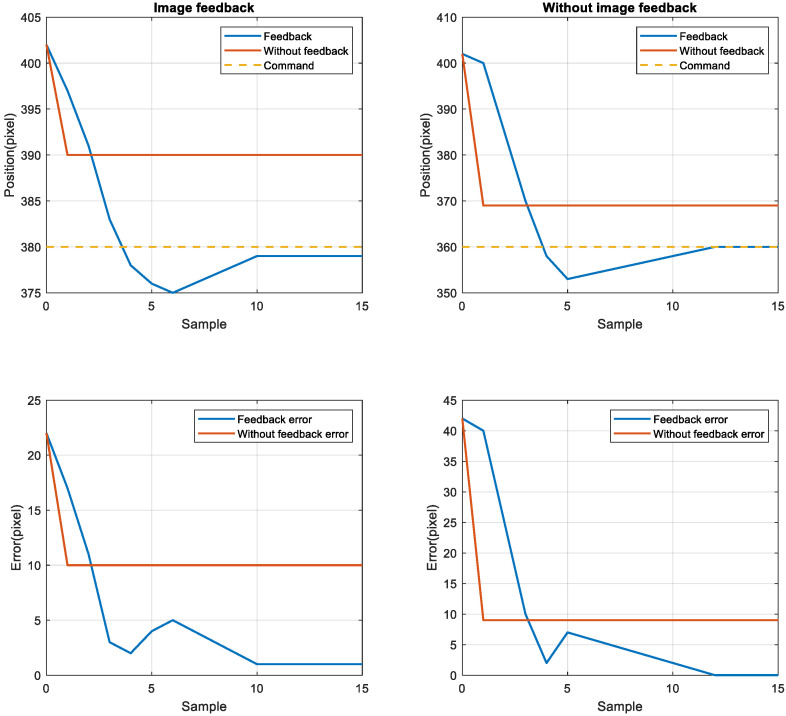
Image feedback system and nonimage image feedback system for positions 380 (**left top** and **bottom**) and 360 (**right top** and **bottom**).

**Figure 13 sensors-23-01938-f013:**
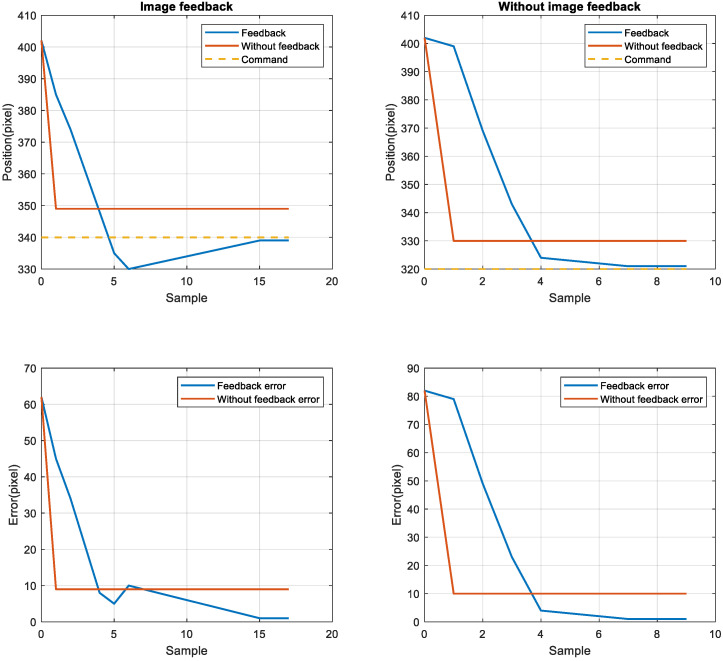
Image feedback system and nonimage image feedback system for positions 340 (**left top** and **bottom**) and 320 (**right top** and **bottom**).

**Figure 14 sensors-23-01938-f014:**
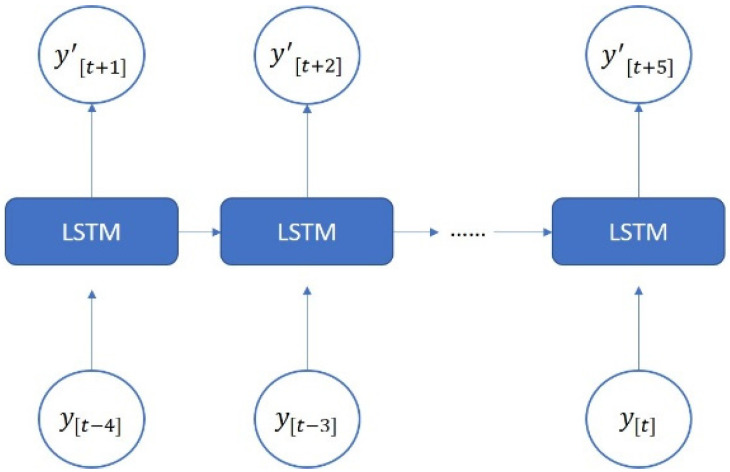
Architecture of a many-to-many LSTM network.

**Figure 15 sensors-23-01938-f015:**
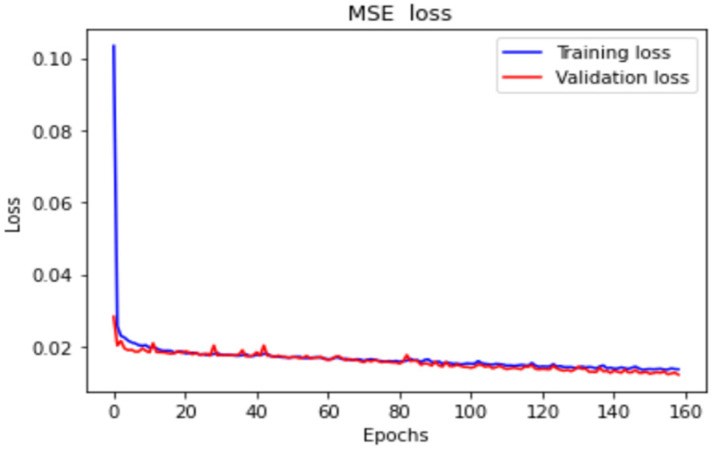
LSTM-based loss function plots.

**Figure 16 sensors-23-01938-f016:**
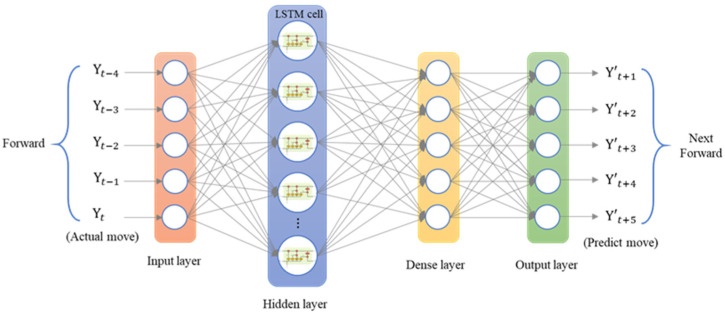
Architecture of an LSTM network for consecutive forward motion.

**Figure 17 sensors-23-01938-f017:**
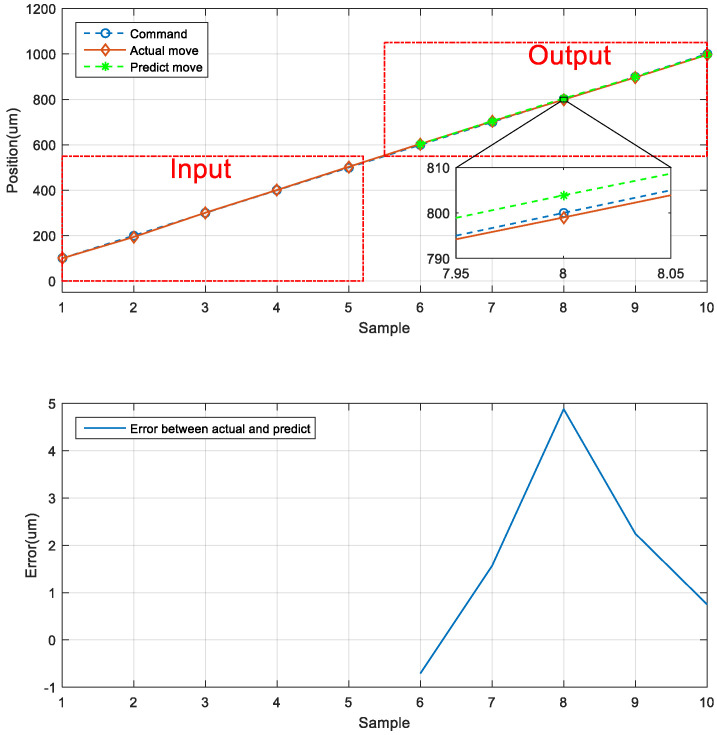
Neural network prediction of the forward position of the stage.

**Figure 18 sensors-23-01938-f018:**
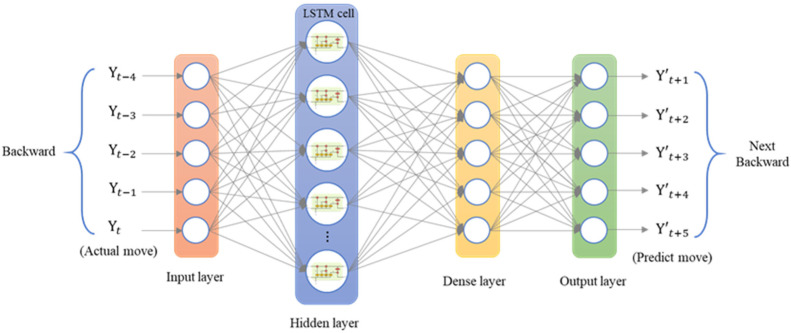
Architecture of an LSTM network for consecutive backward motion.

**Figure 19 sensors-23-01938-f019:**
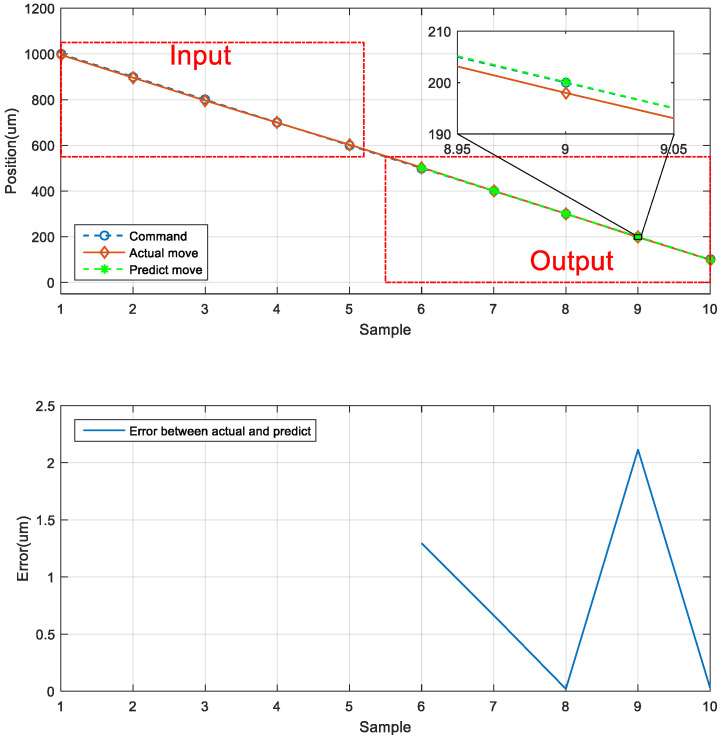
Neural network prediction of the backward position of the stage.

**Figure 20 sensors-23-01938-f020:**
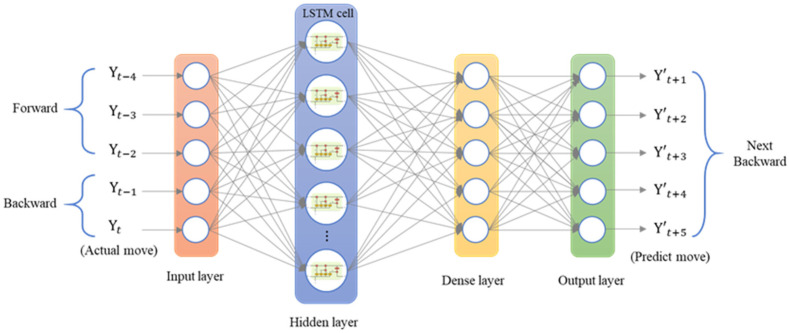
Architecture of an LSTM network for return followed by backward positioning.

**Figure 21 sensors-23-01938-f021:**
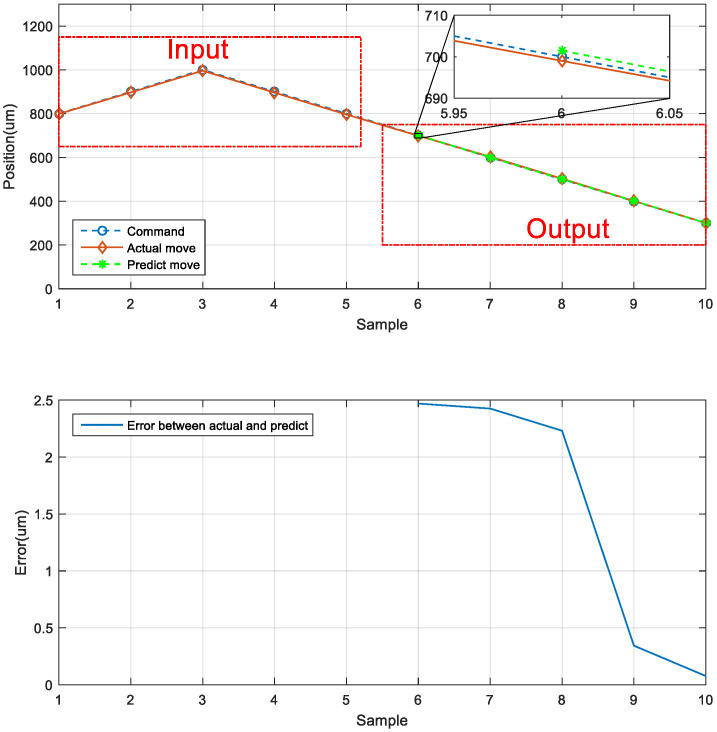
Neural network prediction of forward return position.

**Figure 22 sensors-23-01938-f022:**
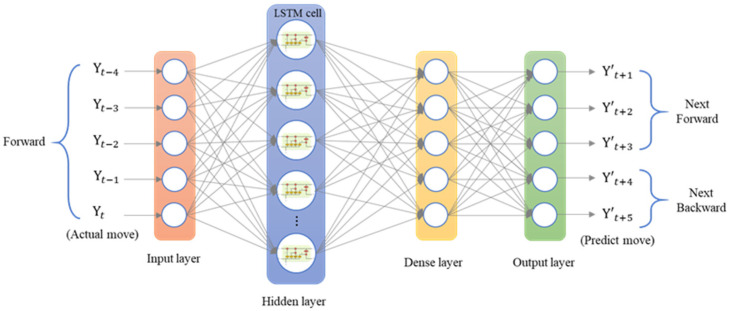
Architecture of an LSTM network for forward followed by return positioning.

**Figure 23 sensors-23-01938-f023:**
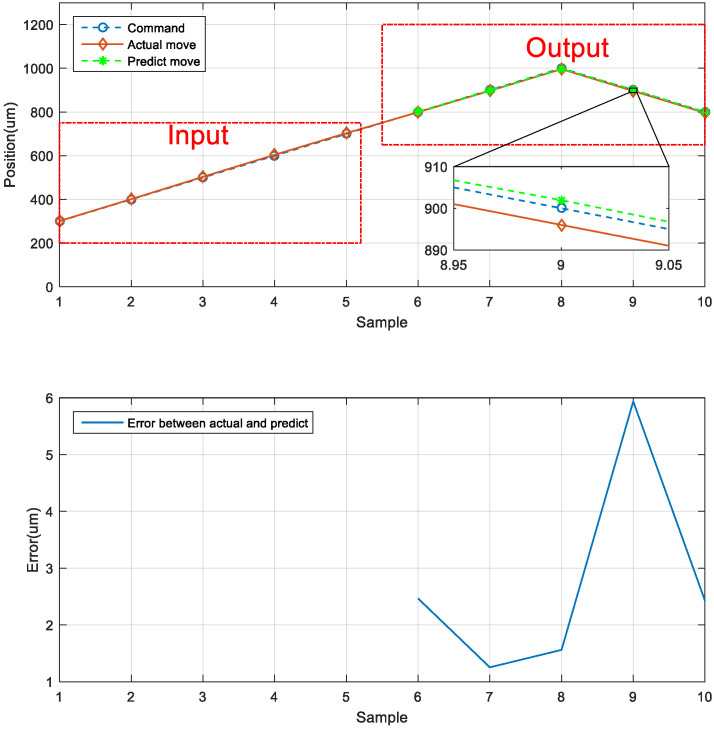
Neural network prediction of backward return position.

**Figure 24 sensors-23-01938-f024:**
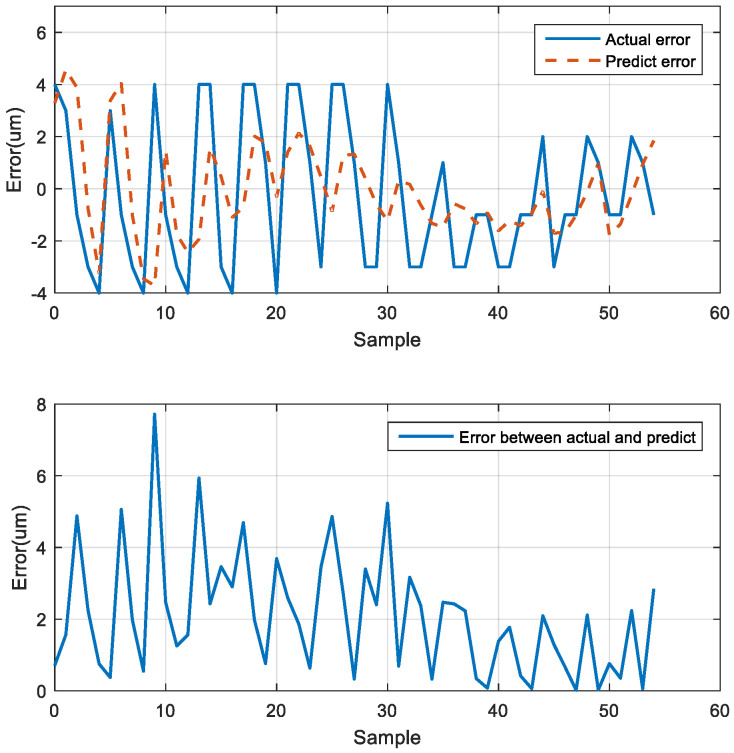
Actual and predicted errors.

**Figure 25 sensors-23-01938-f025:**
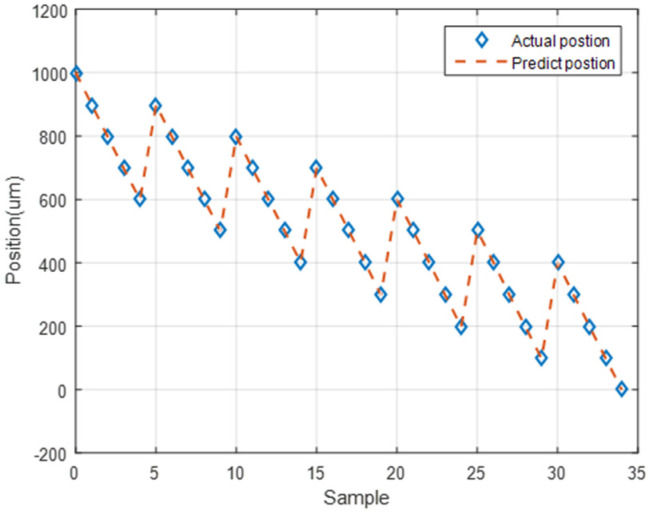
LSTM network predictions and actual positions of the stage during backward stage movement.

**Figure 26 sensors-23-01938-f026:**
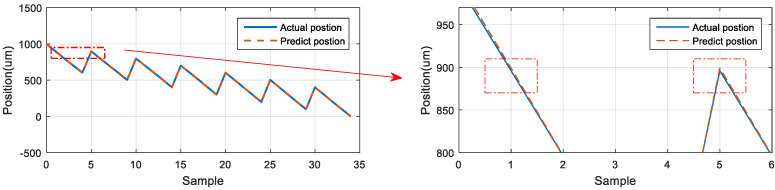
LSTM network predictions and actual positions at 900 μm.

**Figure 27 sensors-23-01938-f027:**
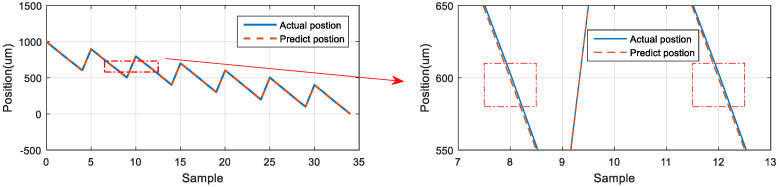
LSTM network predictions and actual positions at 600 μm.

**Figure 28 sensors-23-01938-f028:**
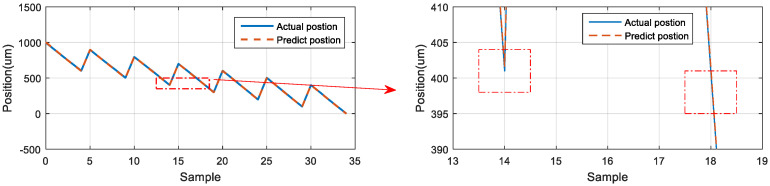
LSTM network predictions and actual positions at 400 μm.

**Table 1 sensors-23-01938-t001:** Specifications of the XXY stage.

Specification	Size
Top Platform (mm)	250 × 250
Bottom Platform (mm)	300 × 300
Height (mm)	78
Load Capacity (Kg)	MAX 30~50
Stroke (mm)	±5
Rotation (Degree)	±3
Center Rotation (Degree)	±5
Body Weight (Kg)	5.5 ± 2%
Parallelism (mm)	±0.03
Motor	AC servo motor
Limit Switch	Photoelectric
Accuracy	8 μm

**Table 2 sensors-23-01938-t002:** Specifications of the CCD equipment.

Specification	Size
Format Size	1/3 type IT Progressive Scan CCD
Effective lines (pixel)	648 × 494
Lens	VS-LD75 0.35 (magnification)
Extension Ring (cm)	40
Scanning Method	Progressive
FOV (mm)	5.4 × 4
Pixel size (μm)	10.75 × 10.48
DOF (mm)	3.4

## Data Availability

Not applicable.

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
