# Peer review of "A Morphing Point-to-Point Displacement Control Based on Long Short-Term Memory for a Coplanar XXY Stage"

_sensors, 2023, doi:10.3390/s23041938_

Round 1

Reviewer 1 Report

In this paper, a visual recognition with a CCD image feedback control was used to feedback the  XXY stage’s movement and the LSTM deep learning model is used to compensate the tracking error. The experimental results indicated an LSTM model used for various types of motion positioning prediction. However, there are some problems which should be clarified. 

1. In Figs. 12 and 13, in the bottom figures, the feedback errors are wrong, because the distances between the command and the feedback in the top figure are different from the bottom figures. Please check them and correct the bottom figures in Figs. 12 and 13. 

2. In Fig. 20, the predict errors are lagger than the actual errors. Therefore, the errors does not converge in the experiments.  Please find a method to deal with this phenomenon.

3. In Section 5.4~5.6, the authors show the prediction of Neural Network. However, the authors did not provide what type of neural networks used in these sections. Please provide the details in the revised one.

4. In Section 5.8, the authors show the LSTM Network Prediction; however, the details of LSTM network are not provided. 

Author Response

Dear reviewer: Please see the attachment. Thanks 

Reviewer 2 Report

The study submitted by the authors is a consolidated work. The system and processes described are logically structured. Figures and tables aid understanding.

What I think needs to be improved is the number of references used, I think that at this level there should be much more references in the paper (13 is not enough). I suggest that this should be done in the following chapters: Introduction, Image acquisition system or LSTM network. Even at the expense of general process descriptions. 

Author Response

(The authors gave the same response as above.)

Reviewer 3 Report

The paper proposes a visual recognition with a charge-coupled device (CCD) image feedback  control system was used to record an coplanar XXY stage’s movement, however, is necessary to clarify some process used to perform this analysis. Also, some factual clarifications will be helpful as highlighted in the comments below.

- What is the tolerance range required for this process? And how is defined the high precision on the measurements?

-What is the sample rate of the measurements?

-Additional to Figure 14. Please include the diagram of the Network used? And include its characteristics.

-Is possible to use this proposal for a 3D movement?

-To reach the precision, there are mechanical restrictions?

-The mechanism movement can generate vibrations that affects the measurement?

Author Response

(The authors gave the same response as above.)

Round 2

Reviewer 1 Report

The authors have corrected the original errors in the submission. The paper is fine and publishable.  

Reviewer 2 Report

Dear authors,

I have reviewed the amendments and I find the paper acceptable.

Reviewer 3 Report

The requested changes have been done.